# Grazing and Mowing Affect the Carbon-to-Nitrogen Ratio of Plants by Changing the Soil Available Nitrogen Content and Soil Moisture on the Meadow Steppe, China

**DOI:** 10.3390/plants11030286

**Published:** 2022-01-21

**Authors:** Le Wang, Hengkang Xu, Hao Zhang, Yingjun Zhang

**Affiliations:** 1College of Grassland Science and Technology, China Agricultural University, 2 Yuan Ming Yuan West Road, Haidian District, Beijing 100193, China; wangle202108@126.com (L.W.); xuhengkang@cau.edu.cn (H.X.); zhanghaolyyg2009@163.com (H.Z.); 2Key Laboratory of Grassland Management and Rational Utilization, Ministry of Agriculture and Rural Affairs, Beijing 100193, China

**Keywords:** moderate grazing, grazing exclusion, plant C/N ratios, species-specific response

## Abstract

Common grassland management practices affect plant and soil element stoichiometry, but the primary environmental factors driving variation in plant C/N ratios for different species in different types of grassland management remain poorly understood. We examined the three dominant C/N stoichiometric responses of plants to different land uses (moderate grazing and mowing) in the temperate meadow steppe of northern China. Our results showed that the responses of the C/N ratio of dominant plants differed according to the management practice. The relative abundance of N in plant tissues increased due to increased soil NO_3_^−^, with a consequent decrease in plant C: N in the shoots of *Leymus chinensis*, but the C/N ratio and nitrogen concentration in the shoots of *Bromus inermis* and *Potentilla bifurca* were relatively stable under short-term moderate grazing management. Mowing reduced the concentration of soil NH_4_^+^, thus reducing the nitrogen concentration of the roots, resulting in a decrease in the root C/N ratio of *Potentilla bifurca*. Structural equation model (SEM) showed that the root C/N ratio was affected by both root N and soil inorganic N, while shoot C/N ratio was only affected by the soil inorganic N. Our findings provide a mechanistic understanding of the responses of plant C/N ratio to land use change. The species-level responses of plant stoichiometry to human-managed grasslands deserve more attention.

## 1. Introduction

Grasslands of Inner Mongolia in northern China are productive areas of temperate biodiversity, with great economic and ecological importance [1,2,3]. Moderate grazing and hay-making have been widely demonstrated to be an optimal use of grassland that maintains high plant diversity and productivity [4,5,6]. However, excessively intensive grassland use (e.g., overgrazing) has led to ecosystem degradation of semi-arid grasslands in Northern China over the last several decades [7,8,9]. The process of grassland degradation is largely mediated by the grazing effects on carbon (C) and nitrogen (N) cycling [10,11,12,13]. As a critical element for plants and microorganisms, C and N are linked to biological processes such as N fixation, organic matter decomposition, and plant photosynthesis [4,14].

The change in the plant and soil C/N ratio caused by grazing has attracted much attention because of its influence on the availability of grassland essential nutrients [11,12,13,15]. The C/N ratio affects biogeochemical cycles through effects on ecosystem production and decomposition processes [16]. The plant C/N ratio can indirectly reflect the nutrient absorption and utilization of plants [17]. In addition, the ecophysiological response mechanisms of the dominant species play a key role in an ecosystem’s responses to environmental change [18]. Plant nutrition and nutrient conservation strategies differ between plant species and thus would show different responses to changes in soil resources along the restoration gradient [19]. For example, shallow roots plant production as that mainly rely on fibrous branched roots to acquire surface soil water and nutrients [20]. Therefore, a better understanding of the responses of C/N ratio to grazing and mowing is vital to maintaining grassland functionality and sustainability.

Grazing and mowing strongly affect the quality and quantity of plant detritus returned to the soil from both aboveground and belowground compartments (i.e., shoots and roots) [10]. Grazing directly adds mineral or highly labile organic N to the soil through the return of feces and urine [21]. Thus, plants have the opportunity to absorb more readily available nutrients, thereby lowering the C/N ratio under short-term moderate grazing management. In addition, the aboveground part of the plant has been removed by grazing and mowing, which may promote the uptake of nitrogen from plant leaves [17] and a decrease in shoot C/N. On the contrary, grazing exclusion produces slower soil N mineralization rates and lower soil inorganic N availability [22], which may cause the C/N ratios of the plants to increase. Moreover, grazing and mowing reduces soil moisture and intensifies water limitations for grasslands [23]. Soil water availability has positive effects on soil N transformation and availability in semi-arid grassland [24,25]. However, in the desert steppe, additional water increases N loss and thus decreases N availability, resulting in high N resorption from senescing leaves of *Glycyrrhiza uralensis* [26]. In addition, as moisture limitation increases, grassland nitrogen use efficiency and plant tissue C/N ratios decrease [27,28]. Thus, soil water and nitrogen availability may play an important role in regulating the plant tissue C/N. However, the process by which soil water and nitrogen availability affect shoot C/N and root C/N under different grassland utilization is not clear.

In this study, we examined the effects of management practices (moderate grazing, mowing, grazing exclusion) on the element stoichiometry (C and N) of plant tissues both aboveground and belowground in a meadow steppe in China. The plant community at the site was dominated by the plant species *Leymus chinensis* (Trin.) *Tzvel.*, *Bromus inermis* Leyss. and *Potentilla bifurca* Linn. [29]. The main objectives of this study were (1) to examine whether plant C/N ratios of different dominant species would show different responses to grazing and mowing; (2) to investigate the process by which soil water and nitrogen availability affect the shoot C/N and root C/N under different grassland utilization. We hypothesize that (i) different dominant species had different C/N responses to grazing and mowing due to different nutrient uptake strategies and (ii) grazing and mowing affect the plant C/N ratio of shoot and root by changing the available nitrogen content and soil moisture.

## 2. Results

### 2.1. Dry Weight Proportions of Three Main Plants and Soil Water Content

Dry weight proportions of the three main species changed under different grassland utilization systems (Figure 1a). Compared with the control (CK), the grazing treatment (CG) significantly reduced the dry matter ratios of *Leymus chinensis* and *Bromus inermis* and promoted the growth of forbs (Figure 1a). In addition, moderate grazing significantly reduced soil water content (Figure 1b).

### 2.2. Responses of the Carbon to Nitrogen Ratios of Three Dominant Plants

The effects of grazing and mowing on three dominant plants’ C/N ratios were significantly different (Figure 2). The concentration of TC, TN, and the C/N ratio was relatively stable in the shoots of *Bromus inermis*, but grazing significantly reduced the C/N ratio in the shoots of *Leymus chinensis* by increasing the TN concentration (Figure 2a–c). Grazing increased the TC concentration, but there was no significant change in TN concentration or C/N ratio in the shoots of *Potentilla bifurca* (Figure 2a–c). Mowing significantly reduced the concentration of TC and TN in the roots of *Leymus chinensis* and did not change the root C/N ratio (Figure 2d–f). The C/N ratio in the root of *Potentilla bifurca* was significantly increased in the mowing treatment (Figure 2f). The C/N ratio in the roots of *Leymus chinensis* and *Bromus inermis* were not significantly different under different grassland utilization (Figure 2f).

### 2.3. Soil Nutrient and the C/N Ratio

The total carbon and nitrogen concentration of the soil were unchanged (Figure 3a,b), but the C/N ratio of the soil was significantly reduced by grazing and mowing (Figure 3c). The concentration of ammonium nitrogen (NH_4_^+^-N) in the soil was significantly reduced by grazing and mowing (Figure 3d). Only grazing significantly reduced the concentration of nitrate nitrogen (NO_3_^−^-N) in the soil (Figure 3e). The available nitrogen concentration in the grazing treatment was significantly higher than that in the mowing treatment. There was no significant difference between the grazing exclusion (CK) and the other two treatments (Figure 3f).

### 2.4. Relationship between the C/N Ratio of Three Plants and Environmental Factors

The RDA ordination plot for three plant species indicated that differences in plant and environmental factors explained approximately 94.9%, 87.8%, and 97.9% of the variation in *Leymus chinensis*, *Bromus inermis*, and *Potentilla bifurca*, respectively (Figure 4a–c). In general, the root C/N ratio of *Leymus chinensis* was positively affected by soil NH_4_^+^, soil C/N, and soil water content (Figure 4a). The shoot C/N ratio of *Leymus chinensis* was negatively correlated with soil NO_3_^−^, soil available nitrogen, and shoot N (Figure 4a). In addition, the root C/N ratio of *Bromus inermis* was positively correlated with soil N and negatively correlated with soil NH_4_^+^, soil C/N, and soil water content. The shoot C/N ratio of *Bromus inermis* was positively correlated with soil NO_3_^−^ and soil-available nitrogen and negatively correlated with shoot N (Figure 4b). The root C/N ratio of *Potentilla bifurca* was negatively correlated with soil NH_4_^+^, soil C/N, and soil water content. The shoot C/N ratio of *Potentilla bifurca* was negatively correlated with soil NO_3_^−^, soil available nitrogen, and shoot N (Figure 4c). Structural equation model (SEM) showed that grassland utilization reduced soil moisture and increased soil available nitrogen content. The root C/N ratio was affected by both root N and soil inorganic N, while shoot C/N ratio was only affected by the soil inorganic N (Figure 5).

## 3. Discussion

### 3.1. Driving Factors of Variation in C/N Ratio in Three Plant Species across Different Land Use Types

Moderate grazing significantly reduced the C/N ratio in the shoots of *Leymus chinensis* by increasing the TN concentration in the shoots (Figure 2a–c). Previous studies also suggested that positive grazing induces effects on soil N cycling and leads to greater plant N uptake, which increases concentration of nutrients in plant tissues [30]. Recent studies have shown that leaf nitrogen uptake of *Leymus chinensis* was affected by different grassland utilization modes in a semiarid grassland [17]. However, grazing only increased the shoot nitrogen concentration in *Leymus chinensis*. For *Bromus inermis* and *Potentilla bifurca*, the C/N ratio and TN concentration of shoots was relatively stable. This may be because the nutrient use strategies are significantly different between plant life forms and vegetation types [31]. In addition, mowing significantly reduced the concentration of TC and TN in the roots of *Leymus chinensis* but did not change the root C/N ratio (Figure 2d–f). The C/N ratio in the roots of *Potentilla bifurca* was significantly increased in the mowing treatment (Figure 2f). Our results showed that mowing treatment significantly reduced the concentration of available nitrogen (Figure 3f). This may be due to persistent mowing having had a negative effect on soil nutrient availability through the chronic removal of litter and reduced plant N concentration [32]. Different chemical forms of N in the soil can be utilized by plants (including inorganic N, ammonium (NH_4_^+^) and nitrate (NO_3_^−^), and dissolved organic N (DON)) (Harrison et al., 2007). Different chemical forms of N exhibit different metabolic characteristics during assimilation, leading to different plant preferences for nitrogen [33]. Thus, plant C/N ratios of different dominant species would show different responses to grassland utilization, and different resource utilization strategies of three different species may allow them to coexist. This suggests that plants exhibit plasticity, as illustrated by the trade-off between water use and inorganic nitrogen.

### 3.2. Changes in Soil Nutrient Availability and Soil Moisture in Response to Grassland Management

We observed relatively higher soil NH_4_^+^ and soil moisture in the grazing exclusion areas and higher soil NO_3_^−^ and lower soil moisture under moderate grazing (Figure 1b and Figure 3f). Grazing reduces plant coverage and exposes the soil to air, which increases soil evaporation and in turn reduces soil moisture [34]. Mowing reduces litter input into the soil and further decreases soil carbon and nitrogen availability [35]. Our results also support the finding that soil mineral nitrogen levels were greatest at moderate grazing intensities [36] (Figure 3f). Dung and urine from livestock may also increase soil N availability by enhancing the mineralization rates and modifying some abiotic factors such as soil pH and availability of oxidative organic matter [37]. Previous studies have shown how animal grazing, by enhancing N availability in soils (through dung and urine deposition), can significantly decrease plant above- and belowground C:N ratios [23]. This is consistent with our results (Figure 5a,b).

Grazing and mowing significantly reduced water content (Figure 1b), which restricted the uptake of nitrogen in the underground part of plants (Figure 2b,c), leading to a decrease in root C/N ratio (Figure 5a). Studies have found that there is a significant positive correlation between soil water content and plant root nitrogen content [38,39], which indirectly supports our conclusion. High available N content may directly affect plant C/N ratio by increasing root N content [40] or affecting root dynamics [41]. For example, long-term addition of inorganic nitrogen to grassland soils also significantly reduced the belowground C:N ratio [42]. This indicates that the increase in nitrogen in the soil can directly affect the root C/N stoichiometry. Structural equation model (SEM) also showed that the shoot C/N ratio was indirectly affected by soil inorganic N and indirectly affected by shoot N (Figure 5b). That is probably because grazing increased the available N content and promoted plant aboveground uptake, thus reducing shoot C/N ratio [30]. Thus, the root C/N ratio was affected by both root N and soil inorganic N, while the shoot C/N ratio was only affected by the soil inorganic N.

## 4. Materials and Methods

### 4.1. Study Site

The experiment was conducted at a medium- and long-term grazing experimental site in Hulun Buir, Inner Mongolia, China (49°20′–49°26′ N, 119°55′–120°9′ E) (Figure 6a,b). The elevation of the study area varies from 628 to 649 m. Mean annual temperature is −3–0 °C, and mean annual rainfall is 350–410 mm. The soil at the study site is chernozem or chestnut soil. The grassland was replanted with *Bromus inermis* in 1997 and mowed for hay regularly until the grazing experiment began in 2015. The plant community at the site was dominated by the plant species *Leymus chinensis*, *Bromus inermis*, *Potentilla bifurca*, *Carex pediformis* and *Carex duriuscula* [29]. In this experiment, moderate continuous grazing (CG), grazing exclusion (CK), and mowing treatment (MG) were used (Figure 6c). Each grazing plot covered an area of 1 hectare, with 0.5 hectares for each grazing exclusion and mowed grassland. The sheep grazed in the plot for 90 days annually from June to September. Water and salt bricks were freely available for sheep. The average grazing rate from 2015 to 2018 was 7.53 sheep/ha/90 days. Mowing treatments are often conducted annually in late September, leaving stubble of about 8 cm.

### 4.2. Sampling and Chemical Measurement

The soils were sampled in the middle of September 2019 when the grazing had finished using 27 randomly selected quadrats (0.5 m × 0.5 m) in each plot for a total of 243 quadrats (3 treatments × 3 replicates × 27 quadrats) (Figure 6). *Leymus chinensis* and *Bromus inermis* and *Potentilla bifurca* were the main research plants. The three plant species were collected in each quadrat and cut at the collar to separate the aboveground shoots and the belowground roots. In each quadrat, soil was sampled to a depth of 10 cm with a soil probe (5 cm in diameter). Then, the soil and plant samples taken from the 27 quadrats within a plot were pooled to form one composite plant sample and one composite soil sample for a total of 27 soil samples and 27 plant samples (3 utilization modes × 3 repeats × 3 plant species) (Figure 6). The composite soil samples were divided into two portions by weight; one portion was naturally dried in the shade, and the other portion was stored at −20 °C before further analyses. Soil NH_4_^+^-N and NO_3_^−^-N concentrations were measured using a continuous flow auto-analyzer (Auto-Analyses 3, Seal Analytical, Norderstedt, Germany) [29]. Soil samples at the same time were taken to the laboratory and dried at 105 °C for 48 h to a constant weight to determine soil moisture content. In addition, due to the low reliability of a single moisture measurement, we conducted another soil moisture measurement in November 2019. The naturally dried soil samples were ground to pass through a 0.15 mm mesh sieve, and total C (TC) and total N (TN) were determined. For the analysis of plant carbon and nitrogen concentration, the shoot and root samples were oven-dried to a constant weight at 60 °C for 48 h and then ground to a fine powder to pass through a 0.25 mm sieve. TC and TN were analyzed using an elemental auto-analyzer (Vario MAX CN; Elementar, Hanau, Germany). The aboveground biomass in six quadrants was clipped at ground level in August 2019, as this was considered to be approximately equal to the aboveground net primary production (ANPP) of the current year. The end-of-season biomass was used as an index of ANPP for the nongrazed treatments. A movable exclosure method was used to estimate ANPP in the grazed treatment by using six 1.5 × 1.5 m cages to exclude grazers from the plots for the entire growing season. The growth rate of herbage was calculated by using the caged method in grazing plots.

### 4.3. Statistical Analysis

To analyze the different ways of using grassland with respect to nutrient concentrations and stoichiometric characteristics of soils and plants, one-way ANOVAs was used. The statistical analyses were performed using the SPSS statistical software package (SPSS Version 19.0). To further explore how the carbon to nitrogen ratio of plants and soil were influenced by environment variables, they were analyzed by RDA (Canoco for windows 4.5). Structural equation modeling (SEM) was further used to statistically explore how mechanistic pathways grassland utilization affects the shoot C/N and root C/N of plant (IBM SPSS Amos 21.0.0).

## 5. Conclusions

In summary, our results showed that short-term grassland utilization significantly affected both plant shoot and root C/N stoichiometry. However, plant tissue chemistry may not be a simple reflection of soil nutrient availability, because unpredictable changes in C:N ratios also occurred. Different dominant species had different C/N responses to grazing and mowing due to different nutrient uptake strategies. Grazing and mowing affected the C:N ratios of the shoots and roots of plants by changing the available nitrogen concentration and soil moisture. Our findings provide a mechanistic understanding of the responses of plant C/N ratio to grassland management. The species-level responses of plant stoichiometry to human-managed grasslands and the contribution of different plants to the carbon and nitrogen cycling of ecosystems deserve further study.

## Figures and Tables

**Figure 1 plants-11-00286-f001:**
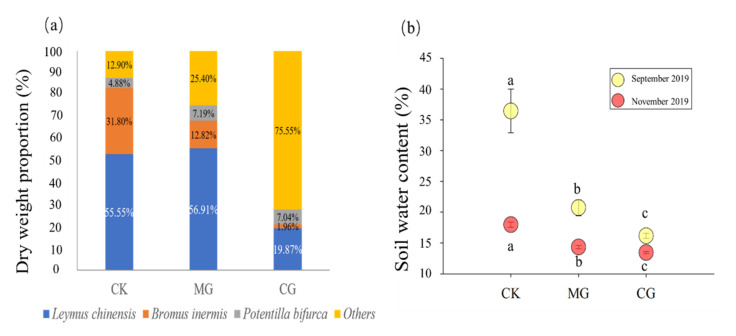
Dry weight proportion of *Leymus chinensis*, *Bromus inermis* and *Potentilla bifurca* (**a**) and the soil water content under (**b**) different grassland utilization modes. CG: continuous grazing; CK: grazing exclusion; MG: mowing grassland. Different letters (a, b and c) above bars indicate significant differences among treatments.

**Figure 2 plants-11-00286-f002:**
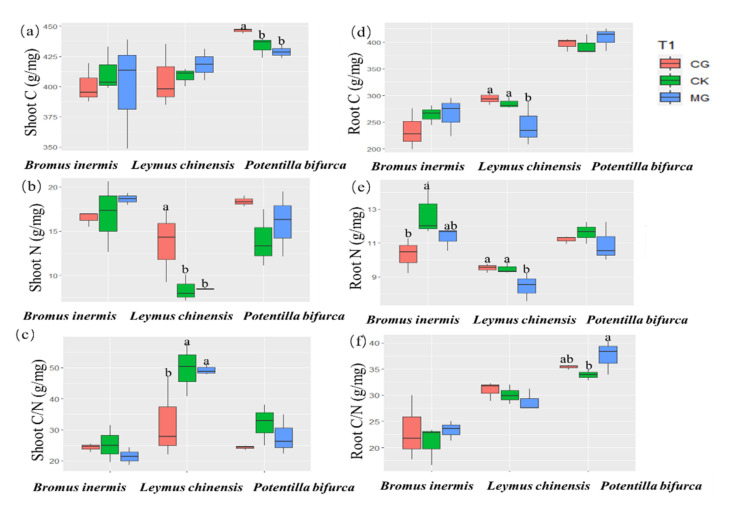
The shoot carbon (**a**), shoot nitrogen (**b**), shoot carbon to nitrogen ratio (**c**), root carbon (**d**), root nitrogen (**e**), and root carbon to nitrogen ratio (**f**) of three different plants under different grassland utilization modes. Note that different letters in the top of the figures indicate significant differences between three grassland utilization modes by the Tukey’s HSD test (N = 3, *p* < 0.05).

**Figure 3 plants-11-00286-f003:**
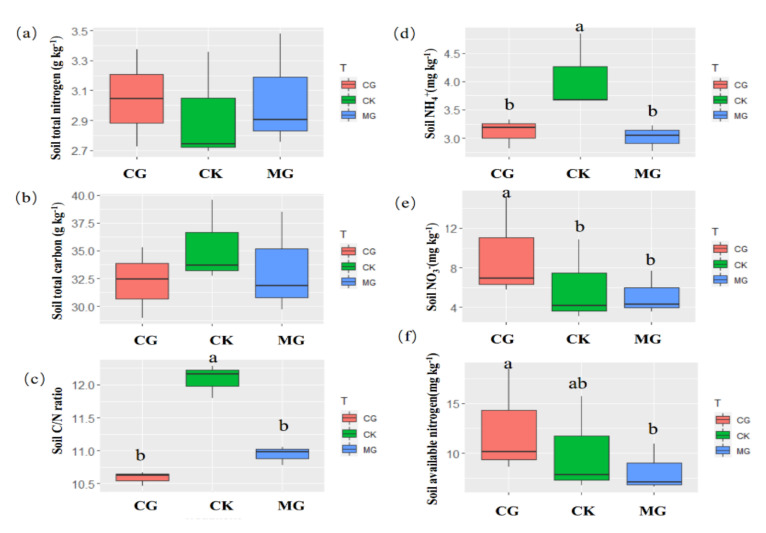
The soil total nitrogen (**a**), soil total carbon (**b**), soil carbon to nitrogen ratio (**c**), soil ammonium nitrogen (**d**), soil nitrate nitrogen (**e**), and soil inorganic nitrogen (**f**) of three different plants under different grassland utilization modes. Different letters at the top of the figures indicate significant differences between three grassland utilization modes by the Tukey’s HSD test (N = 3, *p* < 0.05).

**Figure 4 plants-11-00286-f004:**
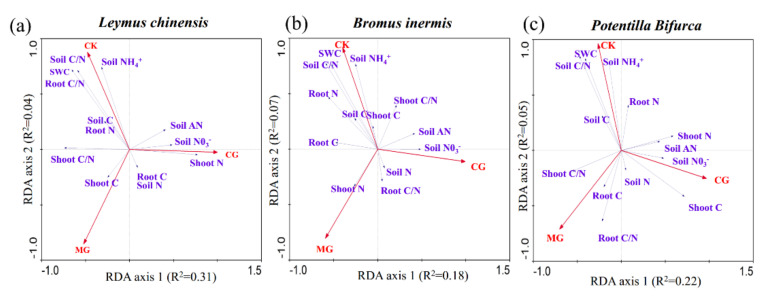
RDA ordination plot for the relationship between the stoichiometry of three different plants and the environment factors in the different grassland utilization modes (**a**–**c**). Soil C: soil total carbon; Soil N: soil total nitrogen; Soil C/N: soil carbon to nitrogen ratio; Soil NH_4_^+^: soil ammonium nitrogen; Soil NO_3_^−^: soil nitrate nitrogen; Soil AN: soil available nitrogen; SWC: soil water content; Shoot C: shoot carbon; Shoot N: shoot nitrogen; Shoot C/N: shoot carbon to nitrogen ratio; Root C: root carbon; Root N: root nitrogen; Root C/N: root carbon to nitrogen ratio; CG: continuous grazing; CK: grazing exclusion; MG: mowing grassland.

**Figure 5 plants-11-00286-f005:**
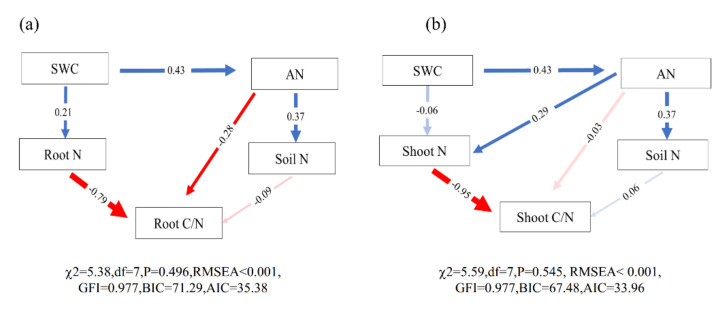
Structural equation models of the direct and indirect effects of soil water content and soil available nitrogen on root C/N (**a**) and shoot C/N (**b**). Soil N: soil total nitrogen; Soil AN: soil available nitrogen; SWC: soil water content. Shoot C/N: shoot carbon to nitrogen ratio; Root C/N: root carbon-to-nitrogen ratio; Shoot N: nitrogen content of shoot; Root N: nitrogen content of shoot. Solid and dashed arrows represent significant (*p* < 0.05) and non-significant (*p* > 0.05) paths. Blue and red arrows represent positive and negative paths. The width of the arrow indicates the strength of the relationship. Numbers adjacent to arrows are standardized path coefficients and are indicative of the effect size of the relationships. The final model fit the data well, as suggested by the chi-square and RMSEA values (**a**), χχ^2^ = 5.38, df = 7, *p* = 0.496, RMSEA < 0.001; (**b**), χχ^2^ = 5.59, df = 7, *p* = 0.545, RMSEA < 0.001.

**Figure 6 plants-11-00286-f006:**
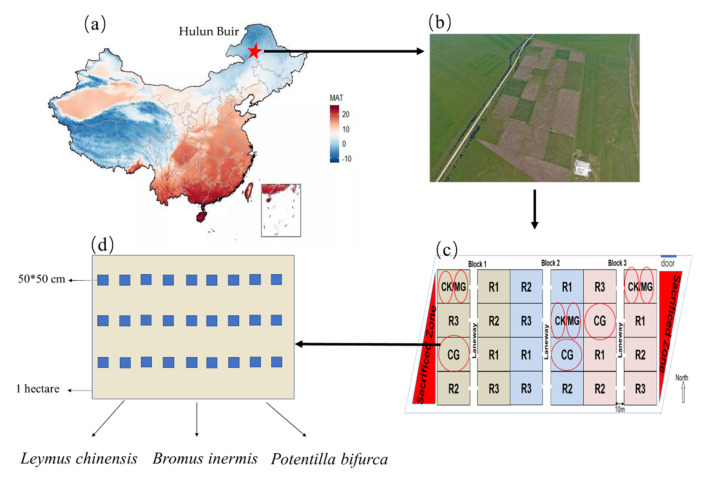
The experiment’s location and a sketch of the sampling mode. The geographical location of the test station (**a**); Aerial photographs of the experiment site (**b**). Sampling involves experi-mental treatment (**c**); Sampling diagram in each treatment (**d**).

## Data Availability

Not applicable.

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
