# Peer review of "Grazing and Mowing Affect the Carbon-to-Nitrogen Ratio of Plants by Changing the Soil Available Nitrogen Content and Soil Moisture on the Meadow Steppe, China"

_plants, 2022, doi:10.3390/plants11030286_

Round 1

Reviewer 1 Report

I have analysed the proposed article and its subject and approach recommend it for publishing after minor revisions.

The article is clear and easy to follow, covering an integrated perspective of the analysed subject. 

The introduction states very well the researched issue as well as the hypothesis and objectives. However, it would be interesting to readdress the hypothesis in the Conclusion section in order to offer a complete overview. 

The results are on point and well structured, while the discussion is detailed enough.  

A few suggestions:

  1. Introduction:

- The introduction contains too few data from specialty literature regarding soil moister from soil in general and from the meadow in particular, as well as its influence on the carbon and nitrogen ratio of different plants. Please add explanations and quotes from literature. Furthermore, other data from literature will be useful (References contains only 34 quotations, a rather few number for such a wide subject).

-Add more details about the plant species and soils from the studied area. The results are rather abrupt, without letting the reader find out more about the specific of the studied area. 

  1. Results:

- figure 1 should contain the explanations of CK, MG, CG (it is true that these are mentioned in the text, but I consider that it is necessary for them to appear on the figure as well in order to understand better the results). Also, the method used to determine soil moisture would fit better at the Methodology section.

- Rows 121-124 should include parentheses (Fig. 5).

- Figure 1b includes % as measure units, while the values are subunits. This must be corrected: 10 instead of 0.10 etc.

  1. Material and Methods:
  • a map with the experiment’s location will be welcomed, as well as a sketch of the sampling mode
  • when the three species are mentioned for the first time, their complete name should be written ex Leymus chinensis (Trin.) Tzvelev

  1. Conclusion: what do you mean by more attention? How can this study be used for further investigations or what areas can be detailed based on it? 

  1. Improve/check the English with a native speaker. Some of the issues I have encountered:

Lines 54-56: Please rephrase. The meaning is unclear. 

Line 56: aboveground part of = the aboveground part of

Line 65: the process by soil = the process by which soil

Line 72: by soil = by which?

Line 266: To analyze the of different = to analyze the different

Author Response

Dear editors and reviewers:

Thank you very much for carefully reviewing the manuscript (plants-1568519) entitled "Grazing and mowing affect the carbon to nitrogen ratio of the plant by changing the soil available nitrogen content and soil moisture on the meadow steppe, China" Suggestions and comments from the reviewers were very helpful in improving our manuscript. Two versions of revised manuscript have been updated as required, one with track changes and a cleaned one. The following is a list of responses to the comments:

Reviewer #1:

I have analysed the proposed article and its subject and approach recommend it for publishing after minor revisions.

The article is clear and easy to follow, covering an integrated perspective of the analysed subject.

The introduction states very well the researched issue as well as the hypothesis and objectives. However, it would be interesting to readdress the hypothesis in the Conclusion section in order to offer a complete overview.

The results are on point and well structured, while the discussion is detailed enough.

Response: Thank you for your helpful and meticulous suggestions, which really helped me a lot. Thank you again for your efforts and hard work!

I've readdressed the hypothesis in the Conclusion section in order to offer a complete overview.

A few suggestions:

  1. Introduction:

- The introduction contains too few data from specialty literature regarding soil moister from soil in general and from the meadow in particular, as well as its influence on the carbon and nitrogen ratio of different plants. Please add explanations and quotes from literature. Furthermore, other data from literature will be useful (References contains only 34 quotations, a rather few number for such a wide subject).

Response: I have added literature and descriptions to enrich the relationship between soil moisture and C/N ratio. (line 62-65)

-Add more details about the plant species and soils from the studied area. The results are rather abrupt, without letting the reader find out more about the specific of the studied area. 

Response: Thank you for your important and useful advice! I have added more details about the plant species from the studied area. (line 72-74)

  1. Results:

- figure 1 should contain the explanations of CK, MG, CG (it is true that these are mentioned in the text, but I consider that it is necessary for them to appear on the figure as well in order to understand better the results). Also, the method used to determine soil moisture would fit better at the Methodology section.

Response: The suggestion has been accepted. The explanations of CK, MG, CG Has been added to the figure 1. Method used to determine soil moisture has been put in the Materials and Methods section.

- Rows 121-124 should include parentheses (Fig. 5).

Response: Fig. 5 has been added. (L147)

- Figure 1b includes % as measure units, while the values are subunits. This must be corrected: 10 instead of 0.10 etc.

Response: The suggestion has been accepted. Figure 1 has been corrected.

  1. Material and Methods:
  • a map with the experiment’s location will be welcomed, as well as a sketch of the sampling mode

Response: Thank you for your important and useful advice! We have added a map with the experiment’s location and a sketch of the sampling mode. (Fig 6)

  • when the three species are mentioned for the first time, their complete name should be written ex Leymus chinensis (Trin.) Tzvelev

Response: The suggestion has been accepted. Three species complete name be written in line 72-73.

  1. Conclusion: what do you mean by more attention? How can this study be used for further investigations or what areas can be detailed based on it? 

Response: We have enriched our conclusions as suggested. (line 282-291)

  1. Improve/check the English with a native speaker. Some of the issues I have encountered:

Lines 54-56: Please rephrase. The meaning is unclear. 

Response: The sentence has been rewritten. (L54-55)

Line 56: aboveground part of = the aboveground part of

Line 65: the process by soil = the process by which soil

Line 72: by soil = by which?

Line 266: To analyze the of different = to analyze the different

Response: These errors have been corrected. Line 56, 67,76, 274

If there is anything unclear, please let us know. We hope the paper now reaches the standards of Plants

Best regards,

Hengkang Xu, Le wang, Yingjun Zhang,

Reviewer 2 Report

Dear Authors

Please find the comments in the attached file

Regards

Author Response

Dear editors and reviewers:

Thank you very much for carefully reviewing the manuscript (plants-1568519) entitled "Grazing and mowing affect the carbon to nitrogen ratio of the plant by changing the soil available nitrogen content and soil moisture on the meadow steppe, China" Suggestions and comments from the reviewers were very helpful in improving our manuscript. Two versions of revised manuscript have been updated as required, one with track changes and a cleaned one. The following is a list of responses to the comments:

Reviewer #2:

According to author guidelines of plants All Figures, Schemes and Tables should be inserted into the main text close to their first citation and must be numbered following their number of appearance (Figure 1, Scheme I, Figure 2, Scheme II, Table 1, etc.). In your case all figures are in one place. It is very unconfortable to read

Response: Thank you very much for your very good advice. All Figures, Schemes and Tables has been inserted into the main text close to their first citation.

The numbers are too small in the figures and almost not readable

This figure is not described in the text in results part. Only in discussion part there are some sentences. Please move this figure to discussion part after it is mentioned or describe it properly in results part

Response: Thank you for your important and useful advice! Figure 1 and figure2 has been enlarged for easy viewing. I would move this figure 5 to discussion part after it is mentioned.

Material and methods- General concern for your results that it lacks the details.

Response: Thank you for your important and useful advice! We have added a map with the experiment’s location and a sketch of the sampling mode. (Fig 6)

In the methods part there is no single word about the Tukey test that you used in results part.

Response: We have changed two-way ANOVAs to One-way ANOVAs was used. (L269).

Please check instructions for authors how the references has to be formated in proper way. Some thext has to be in bold and some parts in italic

Response: We revised the format of the references as required.

If there is anything unclear, please let us know. We hope the paper now reaches the standards of Plants

Best regards,

Hengkang Xu, Le wang, Yingjun Zhang,

Round 2

Reviewer 2 Report

Dear Authors,

Thank you for taking my comments into account.

Regards